# Antibiotic Use Prior to COVID-19 Vaccine Is Associated with Higher Risk of COVID-19 and Adverse Outcomes: A Propensity-Scored Matched Territory-Wide Cohort

**DOI:** 10.3390/vaccines11081341

**Published:** 2023-08-08

**Authors:** Ka Shing Cheung, Vincent K. C. Yan, Lok Ka Lam, Xuxiao Ye, Ivan F. N. Hung, Esther W. Chan, Wai K. Leung

**Affiliations:** 1Department of Medicine, School of Clinical Medicine, The University of Hong Kong, Queen Mary Hospital, Hong Kong; cks634@hku.hk (K.S.C.);; 2Centre for Safe Medication Practice and Research, Department of Pharmacology and Pharmacy, Li Ka Shing Faculty of Medicine, The University of Hong Kong, Hong Kong; vcyan@connect.hku.hk (V.K.C.Y.);; 3Laboratory of Data Discovery for Health (D^2^4H), Hong Kong Science and Technology Park, Hong Kong; 4Department of Pharmacy, The University of Hong Kong-Shenzhen Hospital, Shenzhen 518053, China; 5Shenzhen Institute of Research and Innovation, The University of Hong Kong, Shenzhen 518057, China

**Keywords:** antibiotic, COVID-19 vaccine, adverse outcomes

## Abstract

**Background:** Antibiotics may increase the risk of COVID-19 among non-vaccinated subjects via probable gut dysbiosis. We aimed to investigate whether antibiotics also affect the clinical outcomes of COVID-19 vaccine recipients. **Methods:** This was a territory-wide cohort study of 3,821,302 COVID-19 vaccine recipients (aged ≥ 18 years) with ≥2 doses of either BNT162b2 or CoronaVac. Exclusion criteria included prior COVID-19, prior gastrointestinal surgery, and immunocompromised status. The primary outcome was COVID-19 infection and secondary outcomes included COVID-19-related hospitalization and severe infection (composite of intensive care unit admission, ventilatory support, and/or death). Exposure was pre-vaccination antibiotic use (within 180 days of first vaccine dose). Covariates included age, sex, Charlson Comorbidity Index, and concomitant medication use. Subjects were followed from the index date (first dose vaccination) until outcome occurrence, death, an additional dose of vaccination, or 15 November 2022. Propensity score (PS) matching and a Poisson regression model were used to estimate the adjusted incidence rate ratio (aIRR) of outcomes with antibiotic use. **Results**: Among 342,338 PS matched three-dose vaccine recipients (mean age: 57.4 years; male: 45.1%) with a median follow-up of 13.6 months (IQR: 9.2–16.3), antibiotics were associated with a higher risk of COVID-19 infection (aIRR: 1.16;95% CI: 1.14–1.19), hospitalization (aIRR: 1.75;95% CI: 1.65–1.86), and severe infection (aIRR: 1.60; 95% CI: 1.21–2.11). Notably, antibiotic use was associated with a higher risk of severe infection and death among CoronaVac recipients (aIRR: 1.62 95% CI: 1.18–2.22 and aIRR: 2.70, 95% CI: 1.54–4.73 for the two secondary outcomes, respectively), but not BNT162b2 recipients. **Conclusions:** Pre-vaccination use of antibiotics was associated with a higher risk of COVID-19 infection, hospitalization, and severe disease outcomes.

## 1. Introduction

According to the World Health Organization, COVID-19 has led to nearly 7 million deaths globally as of May 2023. Vaccination is one of the most important measures in preventing severe symptoms and death [1]. There is increasing evidence of the role of the gut microbiome on the impact of COVID-19 disease as well as vaccine immunogenicity [2,3]. Gut microbiota perturbation induced by antibiotics could impair antibody production and affinity among those with low pre-existing antibody titers against influenza virus [4]. Similarly, the early vaccine immunogenicity of BNT162b2 may be impaired by antibiotic usage [5]. The mechanisms underlying these observations include the expression of viral entry receptor angiotensin-converting enzyme 2 (ACE2) [6], the effects of microbiota-derived short chain fatty acids [7], the modulation of B and T cell immune responses [3], and the mediation of the “gut-lung axis”. Gut microbiota alteration by COVID-19 has been associated with inflammatory bowel disease and COVID-19 severity. Intuitively, it has been postulated that gut dysbiosis induced by antibiotics leads to a higher viral load in the gastrointestinal (GI) tract, resulting in a cytokine storm [8] and the impairment of leukocyte function [9]—which is similar to the effects of proton-pump inhibitors (PPIs) in increasing the risk of COVID-19 and severe clinical outcomes in observational studies [10,11].

Worldwide antibiotic use increased by 65% between 2000 and 2015 [12]. Antibiotic misuse was exemplified in a recent study evaluating antibiotic use among 11,047 COVID-19 cases in Hong Kong, carried out between January 2018 to March 2021, that revealed that nearly 30% of cases were given antibiotics—but only less than 2% had confirmed bacterial co-infections [13]. It has been shown that recent usage of antibiotics affects COVID-19 vaccine immunogenicity [5]. However, there is a lack of population-based studies investigating whether antibiotics could increase the risk of COVID-19 and serious clinical outcomes after vaccination. A differential effect may also exist depending on the number of vaccine doses received and the vaccine platform (messenger RNA (mRNA) vs. inactivated virus).

Here, we conducted a territory-wide cohort study on COVID-19 vaccine recipients to determine the effects of antibiotics on the development of COVID-19 and adverse clinical outcomes, with stratification.

## 2. Methods

### 2.1. Data Source

A retrospective cohort study was conducted using the electronic medical record database from the Hospital Authority (HA; including all individuals who ever used HA services), linked with territory-wide vaccination records and COVID-19 confirmed case records from the Department of Health using anonymous unique patient identifiers. Medical history and comorbidities were identified by International Classification of Diseases, Ninth Revision (ICD-9) diagnosis codes, looking back to 2018 at the earliest. Medication use was identified by prescription records using British National Formulary (BNF) codes.

### 2.2. Setting and Study Population

The Hong Kong government provides the public with COVID-19 vaccines (BNT162b2 and CoronaVac) free of charge. Local residents can choose either vaccine according to their own preference. They can receive vaccination according to the recommended vaccination schedule in designated clinics and community vaccination centers.

Individuals aged ≥18 who had received 3 doses of either BNT162b2 or CoronaVac vaccine during the study period (23 February 2021 to 15 November 2022) were included (*n* = 3,299,819). Appendix A shows the study subject selection process. Subsequently, patients with a history of GI surgery (*n* = 17,372), who were immunocompromised (cancer; transplant receipient; primary immunodeficiency; immune-related inflammatory disorders including rheumatoid arthritis, inflammatory bowel disease, and others; splenectomy; end-stage renal disease/dialysis; *n* = 129,984), who had a history of COVID-19 before the first dose of the vaccination (*n* = 18,017), or who used any antibiotic between the first and third dose of vaccination (*n* = 140,060) were excluded. The cohort size of our study was 2,994,386 after exclusion.

Patients were followed from the index date (date of first-dose vaccination) until the earliest of the following: outcome occurrence, death, date of vaccination of an additional dose, or end of data availability (15 November 2022).

### 2.3. Exposures of Interest

The primary exposure of interest is the pre-vaccination use of antibiotics. Appendix A shows the list of antibiotics included and their classification in terms of their anti-aerobic/anti-anaerobic effects and anti-bacterial spectrum. Pre-vaccination antibiotic users were those who received any prescription of any antibiotics within 180 days before their first-dose vaccination [5]. Antibiotic non-users were those who did not receive any prescription of antibiotics within 180 days from the first dose of vaccination. Figure 1 shows the study timeframe.

### 2.4. Outcomes of Interest

Outcomes of interest included: (i) COVID-19, defined by a positive polymerase chain reaction (PCR) test (voluntary reporting of rapid antigen test (RAT) positive test was not included) as confirmed by the Department of Health of Hong Kong Special Administrative Region Government; (ii) COVID-19-related hospitalization, defined as hospital admission within 28 days following a positive PCR test; (iii) COVID-19-related mortality, defined as all-cause mortality within 28 days following a positive PCR test; (iv) severe COVID-19, defined as a composite of COVID-19-related mortality or intensive care unit (ICU) admission or ventilatory support (ICD-9 procedure codes: 39.65, 89.18, 93.90, 93.95, 93.96, 96.7, 96.04) within 28 days following a positive PCR test.

#### 2.4.1. Statistical Analyses

R software (version 3.2.3) was used for all statistical analyses. Continuous variables were expressed as mean with standard deviation (SD) or median with interquartile range (IQR). Discrete variables were expressed as counts with percentage.

Covariates included age, sex, Charlson Comorbidity Index (CCI) score [14], index date, presence of comorbidities (hypertension [15], diabetes mellitus [16], dyslipidemia [17], cardiovascular disease [18], respiratory disease, obesity diagnosis [19], smoking [20], alcohol use disorders [21], ulcers, moderate-to-severe liver disease [22,23,24], chronic renal failure [25]), and medication use within the past 90 days (angiotensin-converting enzyme inhibitors (ACEIs), angiotensin receptor blockers (ARBs) [26], metformin, lipid-modifying drugs, antiplatelets, nonsteroidal anti-inflammatory drugs (NSAIDs), oral anticoagulants, oral corticosteroids, antidepressants, antiviral drugs, PPIs [10,11], or H2 receptor antagonists (H2RAs) [27].

Propensity-score matching was conducted to minimize potential confounding between pre-vaccination antibiotic users versus non-users at a 1:1 ratio using the nearest-neighbor algorithm with a caliper of 0.2, where the propensity score was estimated using logistic regression for the covariates listed above. A standardized mean difference (SMD) of 0.2 or less was considered negligible.

Poisson regression adjusted for covariates was used to estimate the incidence rate ratio (IRR) of outcomes among pre-vaccination antibiotic users versus non-users in the propensity-score (PS)-matched cohort. Stratified analyses by age (<60 and ≥60), sex, Charlson Comorbidity Index categories (0 and ≥1), and vaccine platform were conducted. Effects of the nature of various antibiotics (anti-aerobic vs. anti-anaerobic, narrow- vs. broad-spectrum, and intravenous vs. oral) and different antibiotic classes used pre-vaccination on vaccine effectiveness were also estimated.

To increase the robustness of the study results by demonstrating a biological gradient, the association between pre-vaccination cumulative antibiotic exposure and outcomes were estimated using Poisson regression adjusted for the covariates listed above. Cumulative antibiotic exposure in the past 180 days before first dose vaccination were categorized by their cumulative duration (0 (non-users), 1–7, and ≥8 days). Incidence rate ratios (IRRs) and 95% confidence intervals were reported.

A two-sided *p*-value of less than 0.05 was considered statistically significant.

#### 2.4.2. Statements of Ethics

The study protocol conformed to the ethical guidelines of the World Medical Association Declaration of Helsinki and was approved by the Institutional Review Board of the University of Hong Kong and the Hong Kong West Cluster of Hospital Authority.

## 3. Results

### 3.1. Patient Characteristics

In total, 342,338 three-dose vaccine recipients (171,169 antibiotic users and 171,169 antibiotic non-users) were PS matched. The mean age was 57.2 years (SD: 18.2) in antibiotic non-users and 57.6 (SD: 19.1) in antibiotic users. The median follow-up was 13.6 months (IQR: 9.2–16.3), and the median duration of pre-vaccination antibiotic use was 7.0 days (IQR: 6.0–9.0). The difference in baseline characteristics between antibiotic users and non-users before PS matching and after PS matching (all with SMD < 0.2) is shown in Appendix A and Table 1, respectively.

### 3.2. Association between Pre-Vaccination Antibiotic Use and COVID-19 Outcomes

Among the PS-matched vaccine recipients, 45,173 (13.2%) developed COVID-19. The median time from the index date to development of infection was 7.9 months (IQR: 5.7–10.8), respectively. Pre-vaccination antibiotic use was associated with a higher risk of COVID-19 (aIRR: 1.16; 95% CI: 1.14–1.18), COVID-19-related hospitalization (aIRR: 1.75; 95% CI: 1.65–1.86), and severe disease outcomes (composite of ICU admission/ventilatory support/death—aIRR: 1.60; 95% CI: 1.21–2.11) and death (aIRR: 2.56; 95% CI: 1.56–4.20) (Table 2).

Antibiotic use was associated with a similarly higher risk of infection regardless of the duration of use (Table 3). A biological gradient existed for COVID-19 hospitalization and severe COVID-19. Compared with antibiotic non-use, the use of antibiotics for 7 days or less was associated with an approximately 1.5-fold higher risk of these outcomes (hospitalization aIRR: 1.57 (95% CI: 1.47–1.69); severe outcomes aIRR: 1.49 (95% CI: 1.08–2.04), respectively); while use of antibiotics for ≥8 days was associated with a 1.8- to 2.0-fold higher risk of these outcomes (hospitalization aIRR: 2.07 (95% CI: 1.92–2.23); severe outcomes aIRR: 1.77 (95% CI: 1.26–2.50), respectively; Table 3). However, a longer duration of antibiotic use did not pose a higher risk of COVID-19 mortality compared with shorter durations.

### 3.3. Stratified Analysis

#### 3.3.1. Stratified by Vaccine Platform, Age, Sex, and Charlson Comorbidity Index

The magnitude of the higher infection risk and hospitalization associated with antibiotic use was similar regardless of vaccine platform, age, sex, or CCI (Table 4).

However, antibiotic use was associated with a higher risk of severe disease outcomes and death with CoronaVac only (aIRR: 1.62 95% CI: 1.18–2.22; aIRR: 2.70, 95% CI: 1.54–4.73 for the two secondary outcomes, respectively), aged ≥60 years (aIRR: 1.57 95% CI: 1.17–2.11; aIRR: 2.53, 95% CI: 1.54–4.15 for the two secondary outcomes, respectively), and those with CCI > 1 (aIRR: 1.94 95% CI: 1.38–2.74; aIRR: 2.66, 95% CI: 1.49–4.76 for the two secondary outcomes, respectively). On the other hand, the magnitude of risk for the secondary outcomes was similar for males and females.

#### 3.3.2. Stratified by Nature and Class of Antibiotics

Compared with antibiotic non-use, anti-anaerobic, broad-spectrum, and oral antibiotics were associated with a higher infection risk (aIRR: 1.16, 95% CI: 1.14–1.19; aIRR: 1.16, 95% CI: 1.14–1.18; aIRR: 1.16, 95% CI: 1.14–1.18, respectively), hospitalization (aIRR: 1.71, 95% CI: 1.60–1.83; aIRR: 1.77, 95% CI: 1.67–1.89; aIRR: 1.66, 95% CI: 1.60–1.77, respectively) and severe infection outcomes (aIRR: 1.70, 95% CI: 1.26–2.29; aIRR: 1.68, 95% CI: 1.26–2.22; aIRR: 1.53, 95% CI: 1.14–2.04, respectively; Table 5). Broad-spectrum antibiotics were associated with a higher infection risk (aIRR: 1.16; 95% CI: 1.14–1.18) and severe infection outcomes (aIRR: 1.67; 95% CI: 1.26–2.22). Anti-aerobic, narrow-spectrum, and intravenous antibiotics were associated with a higher infection risk (aIRR: 1.11, 95% CI: 1.07–1.15; aIRR: 1.14, 95% CI: 1.09–1.20; aIRR: 1.15, 95% CI: 1.02–1.29, respectively), but not severe infection outcomes (aIRR: 1.49, 95% CI: 0.88–2.53; aIRR: 0.79, 95% CI: 0.25–2.51; aIRR: 1.92, 95% CI: 0.77–4.76, respectively).

Regarding antibiotic class (Appendix A), compared with antibiotic non-use, penicillins (aIRR: 1.17; 95% CI: 1.15–1.19), cephalosporins (aIRR: 1.12; 95% CI: 1.04–1.21), macrolides (aIRR: 1.12; 95% CI: 1.05–1.19), carbapenems (aIRR: 1.35; 95% CI: 1.16–1.58), quinolones (aIRR: 1.16; 95% CI: 1.11–1.21), tetracyclines (aIRR: 1.20; 95% CI: 1.13–1.28), and aminoglycosides (aIRR: 1.45; 95% CI: 1.11–1.89) were associated with a higher risk of COVID-19—but this was not the case with nitroimidazoles (aIRR: 1.04; 95% CI: 0.98–1.10) or glycopeptides (aIRR: 1.14; 95% CI: 0.92–1.42). All antibiotic classes were associated with a higher risk of COVID-19-related hospitalization. Penicillins and quinolones were associated with a higher risk of severe COVID-19 (aIRR: 1.62, 95% CI: 1.22–2.16 and aIRR: 1.75, 95% CI: 1.06–2.91, respectively).

## 4. Discussion

This is the first study to demonstrate the potential undesirable effects of pre-vaccination antibiotics on the COVID-19 disease course, with a higher risk of infection, hospitalization, and severe disease outcomes being demonstrated in a territory-wide cohort study involving >300,000 PS-matched three-dose vaccine recipients, with a biological gradient being demonstrated. Notably, severe infection and death associated with pre-vaccination antibiotic use was only observed in CoronaVac recipients, but not in BNT162b2 recipients.

Antibiotics can cause significant changes in gut microbiota that have both short- and long-term health consequences [28]. Antibiotics reduce the overall diversity of gut microbiota species—including the loss of some important taxa—which causes metabolic shifts, increases gut susceptibility to colonization, and stimulates the development of bacterial antibiotic resistance [29]. Given the high efficacy of two doses of BNT162b2 and CoronaVac (protection from symptomatic COVID-19 is 95% [30] and 70% [31], respectively), it is hardly surprising that the effects of antibiotics on increasing infection risk was only slightly elevated (5–7%) in two- or three-dose vaccine recipients in the current study.

Although booster doses can enhance antibody production [32,33] and reduce the severity of infection outcomes [34], we showed that antibiotics were still associated with 1.8-fold and 1.6-fold higher risks of hospitalization and severe clinical outcomes (including ICU admission, ventilatory support, and death), respectively, in the three-dose vaccine recipients. Bacterial co-infection causes life-threatening complications in patients with severe viral infections [35], including COVID-19. Studies show that early antibiotic exposure poses risks for childhood asthma, allergies, and airway illnesses [36]. Indiscriminate administration of broad-spectrum antibiotics may increase nosocomial bloodstream infection rates in immunocompromised cancer patients [37]. While the baseline composition of the gut microbiota is mostly restored within 1.5 months, several common species (e.g., *Bifidobacterium adolescentis, Coprococcus eutactus, Coliinsella aerofaciens, Methanobrevibacter smithii, Bididobacterium catenulatum, Bifidobacterium angulatum, and Eubacterium ventriosum*) remain undetectable for at least 180 days [38]. With the combination of the immunocompromising effects of the viral infection and the antibiotic-driven depletion of commensal gut microbes, it has been proven that microorganisms from the dysbiotic gut microbiome translocate into the blood of COVID-19 patients, leading to dangerous blood-streamed infection during COVID-19 [39,40].

Subgroup analysis showed that antibiotic use was associated with a higher risk of adverse outcomes for COVID-19 with CoronaVac, but not BNT162b2, among the three-dose recipients. BNT162b2 mounts a higher neutralizing antibody level of nearly 10-fold more than CoronaVac after two doses of vaccine [40]. Owing to the waning effects of serum antibody levels, vaccine effectiveness against infection progressively diminishes [41]. This corroborates the hypothesis that a higher humoral response induced by BNT162b2 may minimize the unfavorable effects of antibiotics on vaccine immunogenicity and adverse clinical outcomes. A similar phenomenon was observed among older individuals and those with comorbidities, but not among younger age groups and those without comorbidities. The presence of comorbidities is linked to decreased immunity, reduced functional status, and increased healthcare utilization [42]. Advanced age and the number of concurrent comorbidities could also have a synergistic adverse effect on the health status of the elderly, explaining the higher COVID-19 mortality observed among elderly comorbid patients [43].

It was observed that broad-spectrum antibiotics were associated with a higher infection risk in the three-dose vaccine recipients, but not narrow-spectrum antibiotics. Furthermore, the intravenous route of the antibiotics had an overall higher risk of adverse infection outcomes than the oral route. Both gram-positive and gram-negative bacteria are disrupted by broad-spectrum antibiotics, as opposed to narrow spectrum antibiotics [44]. Broad-spectrum antibiotics decrease the Firmicutes to Bacteroidetes (F/B) ratio [45], which is associated with maintaining homeostasis, and changes in this ratio are regarded as dysbiosis. Models have shown that broad-spectrum antimicrobials (specifically vancomycin, metronidazole, and lacticin) cause population shifts from Firmicutes to Proteobacteria in the distal colon. In contrast, the narrow-spectrum bacteriocin (thuricin CD) seems to cause negligible changes in gut flora [46]. One study revealed a significant difference in the antibiotic resistance gene pools in mice treated with tetracycline between intravenous injection and oral administration of the drug, suggesting that drug administration routes influence the level of antibiotic resistance in gut microbiota [47]. The exact mechanism between the type, dosage, duration, administration, and pharmacokinetic and pharmacodynamics properties of antibiotics with vaccine immunogenicity and disease outcomes requires further investigation.

This study had several strengths: First, the population-based study with a large sample size allowed us to detect a statistically significant, albeit small in magnitude, increase in risk of infection—which is important for verifying the hypothesis of antibiotic-induced gut microbiota dysbiosis leading to impaired vaccine immunogenicity. Second, the demonstration of a biological gradient for the majority of outcomes increased the robustness of the study results and strengthened our underlying hypothesis. Third, reverse causality (i.e., symptomatic COVID-19 leading to antibiotic use before infection diagnosis) was minimized, as antibiotic use was defined before the first dose of vaccination while infection outcomes were only counted after the second dose of vaccination. With such a study design, the minimal interval between the commencement of antibiotics was at least three and four weeks for BNT162b2 and CoronaVac recipients, respectively. Fourth, since this was a cohort study, there was less recall and interviewer bias in terms of antibiotics as compared with a case–control study. To minimize the effects of confounding factors on the causality, PS matching ensured the similarity of the baseline characteristics between antibiotic users and non-users in order to reach a quasi-experimental design that may not be otherwise have been achieved by a randomized clinical trial (RCT).

Several limitations of this study should be noted: First, unmeasured confounding may still exist due to the observational study design, although all major confounding factors affecting vaccine immunogenicity and disease outcomes were minimized by the large sample size and PS matching design (Table 1). Second, gut microbiota data was not available; thus, the exact mechanisms of antibiotic-induced gut dysbiosis on vaccine immunogenicity and COVID-19 disease outcomes require further study. Third, asymptomatic infection cannot be ruled out, as this study included only PCR-confirmed positive case, which may overestimate the effects of antibiotics. Fourth, data on further booster doses and bivalent vaccines were not available before the study end date. Fifth, over-the-counter antibiotic usage information was not available, which may bias the study results. For example, self administered antibiotics may be taken incorrectly for too short a time, with wrong doses or timing with food intake. Lastly, COVID-19 vaccination effectiveness may vary in different waves of the epidemic due to the difference in the number of infected cases and prevention protocol/testing criteria from government policies. A prior study showed different frequencies of COVID-19 cases in different waves before the rollout of the vaccination program [48].

## 5. Conclusions

Antibiotics were associated with a higher COVID-19 risk, hospitalization, and severe infection in three-dose vaccine recipients. Further studies are warranted to investigate the interactions between gut microbiota and antibiotics and their effects on the COVID-19 vaccine’s immunogenicity and clinical outcomes. These results further reinforce the importance of the judicious use of antibiotics to minimize their direct and indirect impact on health.

## Figures and Tables

**Figure 1 vaccines-11-01341-f001:**
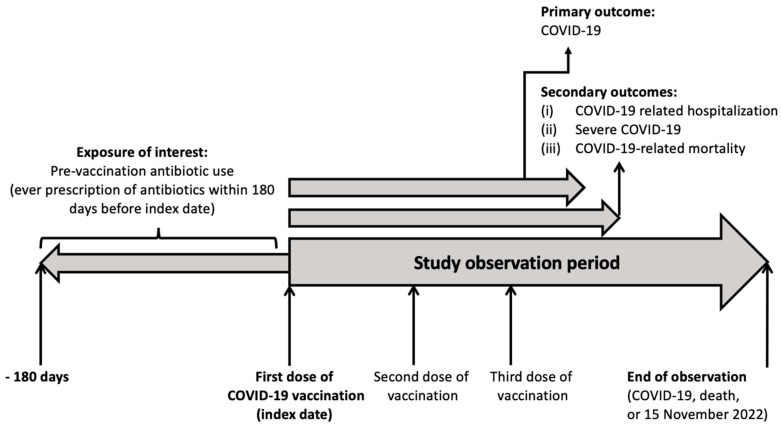
Study timeframe.

**Table 1 vaccines-11-01341-t001:** Baseline characteristics of pre-vaccination antibiotic users and non-users after propensity score matching.

	Pre-Vaccination Antibiotic Non-Users (*n* = 171,169)	Pre-Vaccination Antibiotic Users (*n* = 171,169)	SMD
Age, years (mean (SD))	57.21 (18.2)	57.60 (19.1)	0.021
Sex, male (%)	76,808 (44.9)	77,638 (45.4)	0.010
Charlson Comorbidity Index (mean (SD))	0.40 (0.8)	0.41 (0.8)	0.019
Vaccine platform—CoronaVac (%)	91,576 (53.5)	93,168 (54.4)	0.019
Received 3rd dose (%)	1.00 (0.0)	1.00 (0.0)	<0.001
**Comorbidities—no. (%)**			
Hypertension	51,436 (30.0)	51,294 (30.0)	0.002
Diabetes mellitus	26,124 (15.3)	26,414 (15.4)	0.005
Dyslipidemia	29,851 (17.4)	29,351 (17.1)	0.008
Cardiovascular diseases	57,263 (33.5)	57,375 (33.5)	0.001
Respiratory diseases	8634 (5.0)	9169 (5.4)	0.014
Obesity diagnosis	8377 (4.9)	8626 (5.0)	0.007
Smoking	2100 (1.2)	2279 (1.3)	0.009
Alcohol use disorders	1047 (0.6)	1066 (0.6)	0.001
Ulcers	3569 (2.1)	3921 (2.3)	0.014
Moderate-severe liver disease	334 (0.2)	372 (0.2)	0.005
Chronic renal failure	3358 (2.0)	3459 (2.0)	0.004
**Medication use in past 6 months—no. (%)**			
ACEIs	15,317 (8.9)	15,693 (9.2)	0.008
ARBs	18,010 (10.5)	18,074 (10.6)	0.001
Metformin	21,302 (12.4)	21,316 (12.5)	<0.001
Lipid lowering agents	46,982 (27.4)	46,076 (26.9)	0.012
Antiplatelets	23,053 (13.5)	23,134 (13.5)	0.001
NSAIDs	28,861 (16.9)	26,385 (15.4)	0.039
Oral anticoagulants	4739 (2.8)	4897 (2.9)	0.006
Steroids	3478 (2.0)	4104 (2.4)	0.025
Antidepressants	12,873 (7.5)	12,939 (7.6)	0.001
Antiviral drugs	3214 (1.9)	3231 (1.9)	0.001
PPIs	36,374 (21.3)	36,571 (21.4)	0.003
H2RAs	41,112 (24.0)	38,394 (22.4)	0.038

SMD: standardized mean difference, SD: standard deviation, ACEIs: angiotensin-converting enzyme inhibitors, ARBs: angiotensin receptor blockers, NSAIDs: non-steroidal anti-inflammatory drugs, PPIs: proton pump inhibitors, H2RAs: H2 receptor antagonists.

**Table 2 vaccines-11-01341-t002:** Association between pre-vaccination antibiotic use and COVID-19 outcomes after three doses of BNT162b2/CoronaVac.

	Events	Person-Days	No. of Persons	Incidence Rate (per 100,000 Person-Days)	Adjusted IRR (95% CI)
*COVID-19*					
Never users	21,110	66,365,213	171,169	31.80883	-
Pre-vaccination antibiotics users	24,063	65,191,587	171,169	36.9112	1.16 (1.14–1.18)
*COVID-19-related hospitalization*					
Never users	1550	70,426,545	171,169	2.200875	-
Pre-vaccination antibiotics users	2889	69,621,220	171,169	4.149597	1.75 (1.65–1.86)
*Severe COVID-19 (ICU admission/ventilatory support/death)*					
Never users	78	70,655,136	171,169	0.110395	-
Pre-vaccination antibiotics users	138	70,067,227	171,169	0.196954	1.60 (1.21–2.11)
*COVID-19-related death*					
Never users	21	70,665,654	171,169	0.029717	-
Pre-vaccination antibiotics users	66	70,080,467	171,169	0.094177	2.56 (1.56–4.20)

IRR: incidence rate ratio; CI: confidence interval.

**Table 3 vaccines-11-01341-t003:** Association between cumulative duration of pre-vaccination antibiotic exposure and COVID-19, hospitalization and severe COVID-19.

Cumulative Duration (within 180 Days Pre-Vaccination)	Events	Person-Days	Persons	Incidence Rate (per 100,000 Person-Days)	Adjusted IRR (95% CI)
COVID-19 infection					
0 days (non-users)	21,110	66,365,213	171,169	31.80883	-
1–7 days	16,911	47,596,958	122,112	35.52958	1.140 (1.117–1.164)
≥8 days	7152	17,594,629	49,057	40.64877	1.212 (1.180–1.246)
COVID-19 hospitalisation					
0 days (non-users)	1550	70,426,545	171,169	2.200875	-
1–7 days	1641	50,791,737	122,112	3.23084	1.572 (1.466–1.685)
≥8 days	1248	18,829,483	49,057	6.627904	2.070 (1.919–2.232)
Severe COVID-19					
0 days (non-users)	78	70,655,136	171,169	0.110395	-
1–7 days	78	51,044,140	122,112	0.152809	1.485 (1.083–2.035)
≥8 days	60	19,023,087	49,057	0.315406	1.772 (1.258–2.496)
COVID-19 mortality					
0 days (non-users)	21	70,665,654	171,169	0.029717	-
1–7 days	40	51,051,180	122,112	0.078353	2.626 (1.543–4.469)
≥8 days	26	19,029,287	49,057	0.136631	2.459 (1.373–4.403)

IRR: incidence rate ratio; CI: confidence interval.

**Table 4 vaccines-11-01341-t004:** Association between pre-vaccination antibiotic use and COVID-19 outcomes, stratified by vaccine platform, age, sex, and Charlson Comorbidity Index.

Pre-Vaccination Antibiotic Use	Adjusted Incidence Rate Ratio (95% CI)							
	BNT162b2	CoronaVac	Age < 60	Age ≥ 60	Male	Female	CCI 0	CCI ≥ 1
*COVID-19*								
	1.17 (1.13–1.20)	1.15 (1.13–1.18)	1.15 (1.12–1.18)	1.18 (1.15–1.21)	1.15 (1.12–1.19)	1.17 (1.14–1.20)	1.14 (1.12–1.16)	1.22 (1.17–1.26)
*COVID-19-related hospitalization*								
	1.69 (1.48–1.92)	1.76 (1.64–1.89)	1.63 (1.40–1.89)	1.76 (1.64–1.88)	1.74 (1.60–1.90)	1.76 (1.61–1.93)	1.73 (1.57–1.90)	1.75 (1.61–1.89)
*Severe COVID-19*								
	1.50 (0.83–2.72)	1.62 (1.18–2.22)	1.80 (0.71–4.54)	1.57 (1.17–2.11)	1.65 (1.14–2.38)	1.56 (1.01–2.40)	0.98 (0.59–1.62)	1.94 (1.38–2.74)
*COVID-19-related mortality*								
	2.02 (0.70–5.85)	2.70 (1.54–4.73)	-	2.53 (1.54–4.15)	2.71 (1.41–5.22)	2.30 (1.07–4.90)	2.18 (0.84–5.64)	2.66 (1.49–4.76)

CI: confidence interval; CCI: Charlson Comorbidity Index.

**Table 5 vaccines-11-01341-t005:** Association between nature of the antibiotics and COVID-19 outcomes after three doses of BNT162b2/CoronaVac.

	Events	Person-Days	No. of Persons	Incidence Rate (per 100,000 Person-Days)	Adjusted IRR (95% CI)
*COVID-19 infection*					
** * Anti-aerobic * vs. *anti-anaerobic* **					
Never users	21,110	66,365,213	171,169	31.80883	-
Anti-aerobic	3527	10,273,829	26,363	34.32995	1.11 (1.07–1.15)
Anti-anaerobic	16,879	45,822,622	119,732	36.83552	1.16 (1.14–1.19)
Both	3657	9,095,136	25,074	40.2083	1.21 (1.16–1.25)
** * Narrow- * vs. *broad-spectrum* **					
Never users	21,110	66,365,213	171,169	31.80883	-
Narrow-spectrum	1588	4,630,583	11,613	34.29374	1.14 (1.09–1.20)
Broad-spectrum	20,276	54,929,426	144,534	36.91282	1.16 (1.14–1.18)
Both	2199	5,631,578	15,022	39.04767	1.19 (1.14–1.24)
** * Intravenous * vs. *oral* **					
Never users	21,110	66,365,213	171,169	31.80883	-
Oral	22,447	61,588,197	159,487	36.44692	1.16 (1.14–1.18)
Intravenous	302	704,960	2359	42.83931	1.15 (1.02–1.29)
Both	1314	2,898,430	9323	45.33489	1.24 (1.17–1.31)
*COVID-19-related hospitalization*					
** * Anti-aerobic * vs. *anti-anaerobic* **					
Never users	1550	70,426,545	171,169	2.200875	-
Anti-aerobic	368	10,926,345	26,363	3.368006	1.65 (1.47–1.85)
Anti-anaerobic	1932	48,955,027	119,732	3.946479	1.71 (1.60–1.83)
Both	589	9,739,848	25,074	6.047322	1.98 (1.80–2.18)
** * Narrow- * vs. *broad-spectrum* **					
Never users	1550	70,426,545	171,169	2.200875	-
Narrow-spectrum	106	4,933,949	11,613	2.148381	1.32 (1.08–1.61)
Broad-spectrum	2516	58,654,252	14,4534	4.289544	1.77 (1.67–1.89)
Both	267	6,033,019	15,022	4.425645	1.77 (1.55–2.02)
** * Intravenous * vs. *oral* **					
Never users	1550	70,426,545	171,169	2.200875	-
Oral	2373	65,784,698	159,487	3.607222	1.66 (1.56–1.77)
Intravenous	86	749,421	2359	11.47553	1.90 (1.53–2.36)
Both	430	3,087,101	9323	13.92893	2.49 (2.23–2.78)
*Severe COVID-19 (ICU admission/ventilatory support/death)*					
** * Anti-aerobic * vs. *anti-anaerobic* **					
Never users	78	70,655,136	171,169	0.110395	-
Anti-aerobic	17	10,982,965	26,363	0.154785	1.49 (0.88–2.53)
Anti-anaerobic	100	49,255,135	119,732	0.203025	1.70 (1.26–2.29)
Both	21	9,829,127	25,074	0.213651	1.28 (0.79–2.08)
** * Narrow- * vs. *broad-spectrum* **					
Never users	78	70,655,136	171,169	0.110395	-
Narrow-spectrum	3	4,950,897	11,613	0.060595	0.79 (0.25–2.51)
Broad-spectrum	125	59,041,017	144,534	0.211717	1.67 (1.26–2.22)
Both	10	6,075,313	15,022	0.164601	1.28 (0.66–2.47)
** * Intravenous * vs. *oral* **					
Never users	78	70,655,136	171,169	0.110395	-
Oral	112	66,147,965	159,487	0.169317	1.53 (1.14–2.04)
Intravenous	5	763,064	2359	0.655253	1.92 (0.77–4.76)
Both	21	3,156,198	9323	0.665357	2.04 (1.25–3.34)
*COVID-19-related death*					
** * Anti-aerobic * vs. *anti-anaerobic* **					
Never users	21	70,665,654	171,169	0.029717	-
Anti-aerobic	7	10,984,950	26,363	0.063724	2.24 (0.95–5.27)
Anti-anaerobic	51	49,263,702	119,732	0.103524	2.89 (1.73–4.83)
Both	8	9,831,815	25,074	0.081368	1.59 (0.70–3.61)
** * Narrow- * vs. *broad-spectrum* **					
Never users	21	70,665,654	171,169	0.029717	-
Narrow-spectrum	2	4,951,046	11,613	0.040396	2.19 (0.51–9.39)
Broad-spectrum	59	59,053,118	144,534	0.09991	2.60 (1.57–4.30)
Both	5	6,076,303	15,022	0.082287	2.31 (0.87–6.15)
** * Intravenous * vs. *oral* **					
Never users	21	70,665,654	171,169	0.029717	-
Oral	53	66,158,741	159,487	0.08011	2.51 (1.51–4.17)
Intravenous	3	763,471	2359	0.392942	3.23 (0.95–10.93)
Both	10	3,158,255	9323	0.316631	2.72 (1.26–5.86)

IRR: incidence rate ratio; CI: confidence interval.

## Data Availability

Data sharing will be available upon reasonable request.

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
