# Peer review of "Antibiotic Use Prior to COVID-19 Vaccine Is Associated with Higher Risk of COVID-19 and Adverse Outcomes: A Propensity-Scored Matched Territory-Wide Cohort"

_vaccines, 2023, doi:10.3390/vaccines11081341_

Round 1

Reviewer 1 Report

The authors collected more than three million samples for this study. This is a well-designed study with convincing data. I would recommend acceptance with minor revision:

1. It is highly suggested that ethical approval should be addressed in the method section.

2. For the flowchart of this study, please indicate how many patients were initially included and how many patients were withdrawing from this study.

3. The inclusion and exclusion criteria should be included in the method section.

Author Response

  1. It is highly suggested that ethical approval should be addressed in the method section.

We thank you for your comments. Statements of ethics were added in the method section (p.4; line155-158)

  1. For the flowchart of this study, please indicate how many patients were initially included and how many patients were withdrawing from this study.

We thank you for your comments. Please refer to p.2 line 92-97 and p.3 line 98-99

  1. The inclusion and exclusion criteria should be included in the method section.

We thank you for your comments. The selection process of the study population was stated under study population in the method section. Patients who were included and excluded were described. (p.2; line 93-97 and p.3 line 98-99) 

Reviewer 2 Report

This is beautiful piece of work, well designed. However, I suggest that the Introduction should be substantially improved. As it is now, some information about the study topic is missing.

Also, please include a list of abbreviations and the study limitations

Some minor spelling errors and long sentences were found. Authors should carefully go through the entire manuscript and effect the corrections

Author Response

We thank you for your comments.

We have revised the introduction and added in extra information. (p. 2 ; line 59-60, 65-68)

List of abbreviations were added. (p. 11 ; line 336-343)

Study limitations were reported in discussion session (p.11; line 315-328) 

We have also proofread the article and made corrections accordingly. Amendments highlighted in red.

Reviewer 3 Report

This is an interesting retrospective study providing valuable information to physicians for their clinical practice not only regarding the pandemic of COVID 19.

Minor revisions need to be addressed

a. Literature addressed in the references can be enriched by additional relevant papers such as Fereira et al 2020, Kaushal and Noor 2022

b. It would be interesting to include information about complications in hospitalized covid 19 patients such as clostridium dificile colitis.

Author Response

  1. Literature addressed in the references can be enriched by additional relevant papers such as Fereira et al 2020, Kaushal and Noor 2022

We thank you for your comments. We have included this paper in the Introduction part: “Gut microbiota alteration from COVID-19 has been associated with inflammatory bowel disease and COVID-19 severity”. (p.2 line 59-60)

Kaushal A, Noor R. Association of Gut Microbiota with Inflammatory Bowel Disease and COVID-19 Severity: A Possible Outcome of the Altered Immune Response. Curr Microbiol. 2022 May 5;79(6):184. doi: 10.1007/s00284-022-02877-7.

  1. It would be interesting to include information about complications in hospitalized covid 19 patients such as clostridium dificile colitis.

We thank you for your comments. As there are various different kinds of complications in hospitalized patients, it will be difficult to list out all the complications, which may not be related to COVID-19 vaccine effectiveness. Instead, we use more related clinical outcomes to indicate severity (e.g. ICU admission, ventilation, mortality), which are standardized outcomes to be reported in COVID-19 clinical research.

Reviewer 4 Report

After reading it, I have the following comments:

1.     All acronyms should be preceded by their full spelling when they first appear in the text (e.g., ACEIs, ARBs, NSAIDs, H2RAS at lines 119-121).

2.     Information should be provided regarding anti-SARS-CoV-2 vaccination campaigns in Hong Kong, especially in terms of how the vaccine is offered, to which subjects (e.g., based on age, occupation, medical history…), and whether it is free or requires payment from recipients.

3.     Information should be provided regarding anti-SARS-CoV-2 vaccination effectiveness in the different waves:

·       doi: 10.3390/jcm11113074

4.     After stating the exclusion criteria, the authors should also state the final study cohort’s size.

5.     The authors should describe which quantitative variables were expressed as means and which were expressed as medians (I suppose the former were continuous quantitative variables, while the latter were discrete ones).

6.     A reference should be provided for the Charlson Comorbidity Index score.

7.     It would be interesting, as a future perspective, to investigate the properness of antibiotics use and antibiotic therapy duration. For instance, it is known that self-administered antibiotic therapies are often taken incorrectly (e.g., for too short a time, with wrong doses or with an incorrect association with food). It is reasonable to assume that subjects who take drugs without medical supervision might be an higher risk of inappropriateness than medically assisted patients, thus accounting for an even higher risk of COVID-19 and its adverse outcomes (as well as other diseases, both infectious and non-communicable). If they see fit, the authors might add these concepts to the article’s discussion.

8.     The paper is very interesting and stems from a fascinating and clever idea. It effectively tackles a pivotal topic such as antibiotics excessive use among the general population. I think this kind of research is much needed, especially in high-income countries where antibiotics are widely employed, and the authors did a commendable job.

 A minor editing of English language should be useful

Round 2

Reviewer 4 Report

Dear Authors

I appreciate your efforts to improve your paper. All my suggestions were include in revised manuscript.